# Analyzing Barriers and Enablers for the Acceptance of Artificial Intelligence Innovations into Radiology Practice: A Scoping Review

Fatma A. Eltawil [1,†], Michael Atalla [1,†], Emily Boulos [1], Afsaneh Amirabadi [2] and Pascal N. Tyrrell [1,3,4,*]

1   Department of Medical Imaging, University of Toronto, Toronto, ON M5S 1A1, Canada; fatma.eltawil@tyrrell4innovation.ca (F.A.E.); m2atalla@uwaterloo.ca (M.A.); emily.boulos@gmail.com (E.B.)
2   Diagnostic Imaging Department, The Hospital for Sick Children, Toronto, ON M5G 1E8, Canada; afsaneh.amirabadi@sickkids.ca
3   Department of Statistical Sciences, University of Toronto, Toronto, ON M5G 1Z5, Canada
4   Institute of Medical Science, University of Toronto, Toronto, ON M5S 1A8, Canada
*   Correspondence: pascal.tyrrell@utoronto.ca; Tel.: +1-416-978-7941
†   These authors contributed equally to this work.

**Abstract:** Objectives: This scoping review was conducted to determine the barriers and enablers associated with the acceptance of artificial intelligence/machine learning (AI/ML)-enabled innovations into radiology practice from a physician's perspective. Methods: A systematic search was performed using Ovid Medline and Embase. Keywords were used to generate refined queries with the inclusion of computer-aided diagnosis, artificial intelligence, and barriers and enablers. Three reviewers assessed the articles, with a fourth reviewer used for disagreements. The risk of bias was mitigated by including both quantitative and qualitative studies. Results: An electronic search from January 2000 to 2023 identified 513 studies. Twelve articles were found to fulfill the inclusion criteria: qualitative studies ($n = 4$), survey studies ($n = 7$), and randomized controlled trials (RCT) ($n = 1$). Among the most common barriers to AI implementation into radiology practice were radiologists' lack of acceptance and trust in AI innovations; a lack of awareness, knowledge, and familiarity with the technology; and perceived threat to the professional autonomy of radiologists. The most important identified AI implementation enablers were high expectations of AI's potential added value; the potential to decrease errors in diagnosis; the potential to increase efficiency when reaching a diagnosis; and the potential to improve the quality of patient care. Conclusions: This scoping review found that few studies have been designed specifically to identify barriers and enablers to the acceptance of AI in radiology practice. The majority of studies have assessed the perception of AI replacing radiologists, rather than other barriers or enablers in the adoption of AI. To comprehensively evaluate the potential advantages and disadvantages of integrating AI innovations into radiology practice, gathering more robust research evidence on stakeholder perspectives and attitudes is essential.

**Keywords:** radiology; radiologist; artificial intelligence; machine learning; computer-aided detection; computer-aided diagnosis

## 1. Introduction

Historically, computer-assisted diagnosis in radiology has primarily relied on computer-aided detection/diagnosis (CAD) systems. CAD functions in a predominantly static manner, employing predefined rules and algorithms to detect or diagnose abnormalities in medical imaging [1]. CAD systems were established to improve exam accuracy, promote a reliable understanding of images, and support related decision-making therapeutic processes [2,3]. They are effective for identifying specific patterns (for example, the detection of suspicious abnormalities on a mammogram), yet they face intrinsic limitations due to their fixed programming and inability to learn from and adapt to new data. This inflexibility

often results in less accuracy in complex or unique cases, limiting their overall utility in diagnostic processes. It is crucial to highlight that the earlier versions of CAD systems, hereafter referred to as "old-CAD", predominantly rely on if–then conditions or traditional machine learning models for generating diagnoses. In contrast, the modern iterations, designated as "new-CAD," have evolved to incorporate contemporary machine learning and/or deep learning techniques and models.

Modern times, however, have seen a shift from traditional methods to the utilization of artificial intelligence (AI), a field that encompasses the development of intelligent systems [1]. These AI systems are capable of tasks that typically necessitate human intelligence, such as natural language understanding, pattern recognition, or decision-making [4]. An essential component of AI is machine learning (ML), which has revolutionized diagnostic methods due to its adaptability and flexibility [2]. ML specifically focuses on creating algorithms and computational models that enable machines to 'learn' from and make accurate predictions based on data, without explicit programming [4]. This capacity for learning and adaptation is in stark contrast to the static nature of CAD, making ML particularly effective for complex and constantly evolving medical diagnostics. The evolution of AI/ML has resulted in dynamic, learning-focused systems that are becoming increasingly proficient and versatile for diagnosing a wider range of conditions, rendering them promising tools in the current medical landscape [2,3]. In the context of this paper, the terms AI and ML are used interchangeably to refer to newer technologies that can learn over time, specifically in medical imaging. Similarly, the term CAD is used interchangeably to encompass both "old-CAD" and "new-CAD", as both systems are considered outdated compared to AI.

AI is one of the fastest-growing branches of informatics and computing and has a high probability of significant influence on healthcare [5–7]. Currently, ML applications are considered to be the most innovative technology for image categorization [8,9]. Fast-developing complex datasets and technologies as well as the increasing needs of radiology departments have placed radiology as a significant candidate for the deployment of AI-based innovations [7]. Specifically, AI-based innovations have increasingly shown the potential to improve triage, diagnosis, and workflow within radiology [10]. Indeed, ML systems have already demonstrated promising results in different fields within radiology, such as the prediction of Alzheimer's disease, mammographic screening, and arthritic joint and muscle tissue segmentation [11–13]. Given the substantial increase in workload in the field of radiology over the last few decades, the utilization of AI and ML presents a valuable opportunity to alleviate this burden [14].

As AI is rapidly moving from a trial phase to an application phase, there has been a concomitant increase in the number of articles about AI in radiology, with an increased rate from 100–150 to 700–800 scientific publications per year during the last decade [15,16]. It is expected that the application of AI in radiology over the next period will progressively advance the quality and depth of the influence of medical imaging on patient care as well as transforming radiologists' workflows [17]. As such, in May 2017, the Canadian Association of Radiologists (CAR) created an AI Working Group with the objective of discussing and focusing on practice, policy, and patient care issues related to the introduction and application of AI in imaging [15]. Furthermore, the 2018 RSNA Artificial Intelligence Summit highlighted that developing systems to deploy ML systems in clinical practice is now an important element in improving algorithm quality and radiology performance [16].

Although there are many anticipated benefits of AI-based innovations in radiology, there are also many barriers to their acceptance, including concerns stemming from radiologists' anxiety regarding possible displacement, uncertainty regarding the acceptance of a novel and, as yet, unknown technology, and the management of complex legal and ethical issues [15,16]. Given the evolving nature of AI applications in radiology, gaining a comprehensive understanding of the obstacles and facilitators surrounding the implementation and successful adoption of these technologies within radiology practice is paramount. This paper aims to shed light on these factors specifically from the viewpoints of physicians or future physicians (i.e., medical students) by reviewing up-to-date literature regarding barriers and enablers associated with the

acceptance of AI-enabled software into radiology practice from a medical provider perspective. There are certainly numerous additional barriers that were not highlighted in the retrieved articles and are therefore beyond the scope of the current review, including practical regulatory limitations, technological shortcomings, and ethical considerations.

## 2. Research Methods and Study Selection

### 2.1. Research Methods

A systematic search was performed using Ovid Medline and Embase (Supplementary Material Tables S1 and S2). PRISMA guidelines were followed for the scoping review The keywords used to generate refined queries were exp diagnosis, computer-assisted/OR exp image interpretation, computer-assisted/OR exp decision making, computer-assisted/exp Radiographic Image Interpretation, Computer-Assisted/Radiologist.mp AND computer-aided diagnosis in radiology.mp., exp Radiographic Image Interpretation, Computer-Assisted/AND exp Radiology/AND exp Artificial Intelligence, exp Artificial Intelligence/ AND exp Radiology/AND exp Decision Making.

### 2.2. Study Selection

The search results were assessed for selection based on article titles and abstracts. Three researchers (F.A.E., M.A., and E.B.) independently evaluated articles to determine their eligibility to be included or not. Any discrepancy between the assessors was resolved by a fourth reviewer (P.N.T.). The article selection process involved the consensus of all researchers to determine if the articles met the predefined inclusion criteria. These criteria included original research or data collection studies written in English, with a focus on physicians or future physicians (i.e., medical students) and the provision of insights into perceived barriers and/or enablers. The reviewers examined the articles to ensure they met these criteria. A stringent set of exclusion criteria was adhered to with the aim of honing the study's focus. As part of our thorough process, we deliberately excluded articles that solely focused on machine learning or deep learning from our predefined keyword search. The reasoning behind this decision can be traced back to the primary research objective: artificial intelligence. Although machine learning and deep learning constitute fundamental parts of artificial intelligence, they do not wholly represent this varied and expansive field. By consciously omitting studies predominantly centered on these specific areas, it was ensured that the investigation maintained a more comprehensive viewpoint, avoiding the pitfalls of becoming excessively specialized or technical. Other exclusion criteria were also applied, which involved guidelines, review articles, meta-analyses, editorials, letters, comments, and conference proceedings, as well as in vitro or animal studies. To ensure a comprehensive examination of the topic, a mixed-methods scoping review approach was adopted, incorporating both quantitative and qualitative articles.

### 2.3. Data Extraction and Analysis

Data obtained from each included study consisted of the publication year, author, title, country of origin, digital object identifier, and journal abbreviation (Table 1: Included Studies).

**Table 1.** Included Studies.

| Study | Geographic Location, Year of Publication | Design of Study | Journal | Barriers Identified | Enablers Identified |
|---|---|---|---|---|---|
| Gong et al. [18] | Canada, 2019 | Survey | Academic Radiology | Anxiety related to displacement" (not "replacement") of radiologists by AI. | ----------------------- |
| Pinto Dos Santos et al. [19] | Europe, 2019 | Survey | European Radiology | Medical students' skepticism about AI providing a definite diagnosis in radiology. | ----------------------- |

**Table 1.** *Cont.*

| Study | Geographic Location, Year of Publication | Design of Study | Journal | Barriers Identified | Enablers Identified |
|---|---|---|---|---|---|
| Strohm et al. [20] | The Netherlands, 2020 | Qualitative | European Radiology | Lack of acceptance and trust of radiologists towards AI, unstructured implementation process. | ---------------------- |
| Lim et al. [21] | Australia, 2022 | Survey | Journal of Medical Imaging and Radiation Oncology | Non-radiologists showed discomfort when acting on AI-generated medical reports. | ---------------------- |
| Povyakalo et al. [22] | United Kingdom, 2013. | Randomized Control Trial | Medical Decision Making | ---------------------- | Improves performance of less-discriminating readers. --> "perception that implementation of a dynamic version of CAD (AI) will decrease errors in diagnosis". |
| Chen et al. [23] | UK, 2021 | Qualitative | BMC Health Services Research | ---------------------- | Radiologists believe AI has the potential to take on more repetitive tasks and allow them to focus on more interesting and challenging work. |
| Alelyani et al. [24] | The Kingdom of Saudi Arabia, 2021 | Survey | Healthcare | ---------------------- | Eighty-two percent of the participants thought that AI must be included in the curriculum of medical and allied health colleges. |
| Huisman et al. [25] | Africa/Europe/North America countries, 2021 | Survey | European Radiology | ---------------------- | Advanced knowledge of AI was inversely associated with the fear of implementation, |
| Lee et al. [26] | United States, 2015 | Focus Group | American Journal of Radiology | Poor acceptance, negative perception of CDS. "Lack of agreement". | Radiologists express a strong desire to be more involved in the implementation of CDS at their respective institutions. "Social influence positive user attitude". |
| van Hoek et al. [27] | Switzerland, 2019 | Survey | European Journal of Radiology | The majority of respondents agreed that AI should be implemented into radiology practice | Students decide against choosing radiology as a residency due to the future of AI in radiology |
| Reeder et al. [28] | United States, 2022 | Survey | Clinical Imaging | AI makes medical diagnosis and Radiologists more efficient | Medical students fear the lack of job opportunities in Radiology due to AI |
| Grimm et al. [29] | United States, 2022 | Mixed-Methods Study | Academic Radiology | Junior medial students held concerns about the limited job opportunities with AI integration | Medical students who have a positive view of AI and senior medical students do not express concerns regarding the scarcity of job opportunities resulting from the integration of AI in radiology. |

## 3. Results Analysis

The studies included in this review were all primary research studies published between January 2000 and January 2023. Out of the 602 articles from our Medline and Embase search (Medline *n*= 224 and Embase = 378), 89 duplicated records were removed before screening (Medline *n* = 21 and Embase *n* = 68). A total of 445 were excluded out of 513 articles based on titles and abstracts. Sixty-eight articles were sought for retrieval, and all were successfully obtained. Of the 68 articles assessed for eligibility, 56 were excluded (32 were review articles; 13 were editorial; 11 were commentaries). Of the initial 513 articles, 12 articles were found to fulfill the inclusion criteria: qualitative studies (*n* = 4), survey studies (*n* = 7), and randomized controlled trials (*n* = 1) (Figure 1).

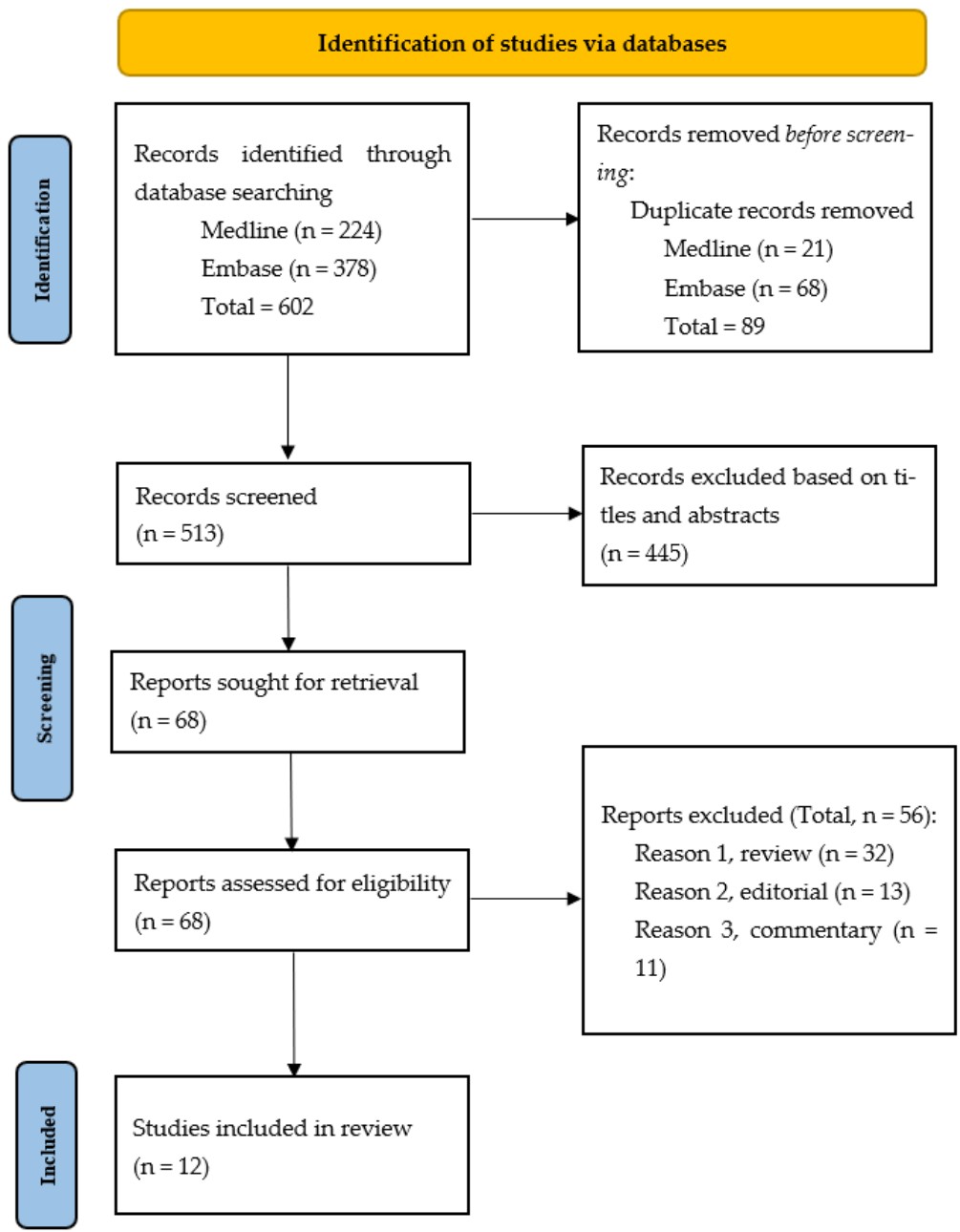

**Figure 1.** Search process.

### 3.1. Articles That Addressed Barriers

A survey study was conducted by Gong et al. to investigate Canadian medical students' perceptions of the impact of AI on radiology and their influences on the students' preference for the radiology specialty [18]. Surveys were distributed to students at all 17 Canadian medical schools with 322 respondents. The results showed that anxiety related to the "displacement" (not "replacement") of radiologists by AI discouraged many medical students from considering radiology as a specialty.

A survey study was conducted by Pinto Dos Santos et al. to determine the attitudes of undergraduate medical students towards AI in radiology and medicine [19]. The surveys were distributed to three major medical schools in Europe, and the anonymity of the students was ensured. A total of 263 medical students responded to the questionnaire. The results of the survey revealed that 56% of the students felt that AI could not provide a definite diagnosis, while 83% disagreed with the notion that AI would replace radiologists. However, the majority (71%) of the students agreed that AI should be integrated into medical education. Interestingly, students who identified as being tech-savvy were more likely (*p*-values between 0.017 and <0.001) to respond yes to whether AI could provide a definite diagnosis.

A qualitative research study was performed by Strohm et al. to recognize barriers to the application of AI in clinical radiology in the Netherlands [20]. The study followed an exploratory qualitative research design, and data collection consisted of 24 semi-structured interviews from seven Dutch hospitals. The study results identified the following AI application barriers: a lack of acceptance and trust of the radiologists towards AI, an unstructured implementation processes, and a lack of empirical evidence on the effect of AI applications on the radiological workflow.

A survey study was conducted by Lim et al. to assess non-radiologists' perceptions of the use of AI in generating diagnostic medical imaging reports [21]. Surveys were distributed from May to August 2021 in tertiary referral hospitals located in Melbourne, Australia with a total of 88 respondents. The results indicated that 35% of the respondents would prefer a radiologist's opinion for an AI report of a simple scan, while 95% of respondents would prefer a radiologist's opinion for an AI report of a complex scan. The study also assessed the comfort level of acting on AI-generated diagnostic medical reports using the Likert scale (0–7). The responses showed that non-radiologists had comfort levels of 6.44 to act based on a radiologist-generated report, 3.57 to act based on an AI-generated report, and 6.38 to act based on a hybrid radiologist/AI-generated report.

### 3.2. Articles That Addressed Enablers

A randomized controlled trial (RCT) study conducted by Povyakalo et al. evaluated how an improved version of CAD, such as AI, would affect the reading performance of radiologists [22]. They established a way to estimate the quality of decisions and identified how computer aids affect it and applied it to the computer-aided detection of cancer by re-analyzing data from a published study where 50 professionals interpreted 180 mammograms, both with and without computer support. In this study, they used stepwise regression to assess the effect of CAD on the probability of a reader making a correct screening decision on a patient with cancer (sensitivity), in this way considering the effects of the difficulty of cancer and the reader's discriminating ability. Regression estimates were used to acquire thresholds for categorizing the cases by difficulty and the readers (by discriminating ability). The results indicated that the use of CAD was associated with a 0.016 increase in sensitivity (95% CI: 0.003, 0.028) for the 44 least-discriminating radiologists for 45 relatively easy, mostly CAD-detected cancers. However, for the six most-discriminating radiologists, the sensitivity decreased by 0.145 with the use of CAD (95% CI: 0.034, 0.257) for the 15 relatively difficult cancers. This study's results show that a dynamic version of CAD (AI) improved the performance of less-discriminating readers, concluding that the implementation of CAD may decrease errors in diagnosis, particularly for less-experienced radiologists.

A qualitative study performed by Chen et al. explored the knowledge, awareness, and attitudes towards AI amongst professional groups in radiology and studied the potential consequences for the future implementation of these technologies into practice [23]. They conducted 18 semi-structured interviews with twelve radiologists and six radiographers from four breast units in National Health Services (NHS) organizations and one focus group with eight radiographers from a fifth NHS breast unit between 2018 and 2020. The study results showed that radiologists believe that AI has the possibility of taking on more monotonous tasks and allowing them to focus on more challenging work. They were less concerned that AI technology might usurp their professional roles and autonomy.

A survey study was conducted in Saudi Arabia by Alelyani et al. to explore the radiology community's attitudes towards the adoption of AI [24]. Data for this study were collected using electronic surveys in 2019 and 2020, and 714 radiologists were included. The results of the survey showed that 82% of the respondents thought that AI must be contained within the curriculum of medical and allied health colleges, 86% of the participants agreed that AI would be important in the future, and 89% of the participants thought that it would never replace radiologists.

A survey study was conducted by Huisman et al. in 54 African, Europe, and North America countries to evaluate the attitudes and fear of radiologists towards the replacement and implementation of AI in radiology [25]. Surveys were distributed through social media, radiological societies, and author networks from April to July 2019, to which there were 1041 respondents. All responses were anonymous, and surveys were translated into nine different languages to ensure accurate responses. Of all the respondents, 21% of the respondents had basic AI knowledge and 16% of participants had advanced knowledge of AI. The results of the survey indicated that 38% of respondents had a fear of AI implementation in their practice, while 48% of respondents showed a proactive and open attitude towards AI implementation. Fear of implementation was reported significantly more often in participants with basic AI knowledge (95% CI: 1.10, 2.21, $p = 0.01$), whereas advanced knowledge of AI was inversely associated with a fear of implementation (95% CI: 0.21, 0.90, $p = 0.03$). Thus, the results of this study indicate that advanced knowledge of AI might enhance the adoption of AI in clinical practice.

### 3.3. Articles That Addressed Barriers and Enablers

A focus group study was conducted by Lee et al. to determine radiologists' insights regarding the application of a clinical decision support system (CDSS) intervention as part of the Medicare Imaging Demonstration project and the influence of decision support on radiologists' interactions [26]. Twenty-six radiologists participated in four focus group discussions. The study results showed poor acceptance and negative perceptions of CDSS as a barrier to AI application. However, one potential enabler identified was the radiologists' expressed desire to have greater involvement in the implementation of CDSS at their respective institutions.

A survey study was conducted by van Hoek et al. to assess the opinions of medical students', radiologists', and surgeons' perceptions of AI incorporation in radiology practice [27]. Surveys were distributed from May to June 2018 to medical institutions within the German-speaking part of Switzerland, to which there were a total of 170 respondents. Responses were recorded using the Likert scale (0–10). In general, respondents showed a positive trend for the use of AI in radiology practice with a mean score of eight. Interestingly, radiologists supported the use of AI in radiology practice significantly more than surgeons ($p = 0.001$) However, a mean score of three was seen with little deviation from all participants when asked whether AI should be used independently for image evaluation after achieving a high diagnostic accuracy. Among the students participating in the survey, 15% stated that they were exploring radiology as a specialization possibility. On the other hand, of those who did not view radiology as a viable specialization choice, 26% mentioned the integration of AI in the field as a factor in their decision.

Reeder et al. conducted a survey study to examine the worry among US medical students about pursuing a career in radiology due to the impact of AI [28]. Surveys were distributed to 32 US medical schools from February to April 2020 with a total of 463 respondents. The results found that 17% of students did not rank radiology as their top specialty due to AI. In addition, 40% of students showed concern about choosing radiology due to AI with 51% of students predicting a decrease in radiology job opportunities due to AI implementation in radiology. However, 77% of students said that they believe that AI will increase the efficiency of radiologists. When students made their specialty selection without the impact of AI, radiology received a significantly higher ranking ($p < 0.0001$), with 21.4% of participants ranking it as their first choice. However, 17% of those who would have chosen radiology as their top pick changed their minds due to the influence of AI.

A mixed-methods study was conducted by Grimm et al. in Vanderbilt University in Nashville, Tennessee to investigate the barriers and stereotypes of medical students towards radiology [29]. From December 2020 to February 2021, surveys were sent to students via email, and 49 participants responded. Of these, 90% agreed (46%) or strongly agreed (44%) that radiologists "get to work with emerging/advanced technology". Next, the students were divided into four focus groups to discuss the various barriers and stereotypes affecting their interest in radiology. During the AI discussion, the focus groups reached a consensus that, while AI would change future workflows, its impact on procedural fields such as interventional radiology would be minimal. Although senior medical students were not apprehensive, second-year medical students expressed concerns about limited job opportunities with AI integration. Furthermore, students who held negative views about radiology raised concerns about the field's long-term sustainability, whereas those with positive views exhibited optimism about AI's contribution. A common consensus within the focus groups was that "AI will supplement but not supplant". The study demonstrated that medical students had a comprehensive understanding of the function of technology, including AI, in radiology and its potential to assist rather than replace diagnostic radiologists with comparatively lesser AI integration in interventional radiology. Additionally, senior medical students and students with positive views showed reduced concerns about the job security of radiologists with the growing integration of AI.

## 4. Discussion

Based on the results of this scoping review, the most common barriers to AI implementation in radiology practice were a lack of awareness, knowledge, trust, and familiarity with the technology, resulting in difficulty for physicians/patients in understanding and accepting it. Other barriers included unstructured implementation processes, a lack of confidence in the benefits of AI being translated into meaningful improvements in patient outcomes, the perceived threat to the professional autonomy of radiologists, and human mistrust of machine-led decisions. Factors that could potentially enable the implementation of AI included the high expectation of AI's likely added value, the potential to decrease errors in diagnosis, the potential to increase efficiency, and the potential to improve the quality of patient care. Interestingly, expectations of AI's added value were indicated as both a barrier (i.e., expectations of low or uncertain clinical potential) and an enabler (i.e., expectations of high clinical potential), highlighting the lack of education and consensus in the field.

Lack of acceptance by radiologists was one of the most important causes of non-adoption and is thus a barrier to the successful implementation of AI, which is consistent with evidence from surveys among radiologists [30,31]. Alexander et al. revealed that many radiologists expressed skepticism about the current diagnostic abilities of AI, mostly in complex patients [32]. Van Hoek et al. found that surgeons were even more skeptical than radiologists with regard to AI in practice [27], and Lim et al. found that non-radiologists had a higher comfort level in acting upon a report generated by a radiologist (with or without assistance from an AI innovation), rather than a report generated by an AI innovation alone [21]. Obermeyer and Emanuel expressed concerns about machines taking jobs away

from humans—reflecting a likely cultural barrier to the implementation of AI in all fields—predicting that "machine learning will displace much of the work of radiologists and anatomic pathologists", and "machine accuracy will soon exceed that of humans" [33]. In another context, anxiety related to the "displacement" of radiologists by AI discouraged many Canadian medical students from considering the radiology specialty [18].

Although some radiologists may feel that AI poses a perceived threat to their professional autonomy in clinical practice, it is more realistic to expect that at least initial implementations of AI will likely follow the model of earlier CAD applications, aiding radiologists and reducing repetitive tasks with a low clinical yield, thereby enabling radiologists to focus on more high-value tasks and work at the top of their license. The successful adoption of these AI tools into the clinical workflow will likely be dependent on radiologists being intimately involved in their development and implementation, as was seen by Lee et al. who noted that radiologists have a strong desire to be involved in the implementation of CDSS in their institutions [26].

To improve the acceptance of AI technology in radiology, education on its role and potential effectiveness could be beneficial for radiologists and referring clinicians, as noted in several of the reviewed studies. While radiologists' perspectives on AI applications range from fear to skepticism to curiosity [20,32], referring clinicians often lack trust in the output of AI applications in radiology. Addressing concerns and providing evidence-based information can demonstrate the potential of AI to increase the diagnostic accuracy and improve patient care [12]. However, the rapid developments in AI technology make it challenging for healthcare professionals to stay current through formal educational avenues alone (for example, in initial training programs). Therefore, incorporating lectures and training sessions on the potential and recent advancements of AI through continuing education sessions can help healthcare professionals to stay up-to-date on specific AI applications in their field and facilitate the widespread acceptance and integration of AI technology into clinical practice. As an additional benefit, including AI education in medical school curricula could allay medical students' concerns when considering a future career in radiology.

Given that the existing literature on the barriers and enablers associated with implementing AI in radiology practice is scant, it is perhaps unsurprising that many radiologists and medical specialists are not aware of these factors, nor how to become more involved in the development and implementation of AI. A study conducted by Ninad et al. highlighted this issue, revealing that over half of the radiology residents surveyed (52%, or 109 residents) expressed interest in AI/ML research but lacked guidance or resources to pursue it [34]. Furthermore, a significant majority of these residents (83%, or 173 residents) agreed that AI/ML should be included in the radiology residency curriculum [34]. These findings show the need for education not only in medical schools but also within radiology residency programs to equip future radiologists with a comprehensive understanding of the potential issues associated with AI implementation.

Encouraging education and mentorship can simultaneously foster radiologists' interest in AI/ML research and improve their competence in integrating these technologies into clinical practice. Given the substantial variation in understanding and interest in AI-based innovations among radiologists and trainees, collaboration between educational institutions and professional bodies is necessary to create structured training programs to establish minimum knowledge standards for radiologists, technologists, and trainees while also offering additional opportunities and incentives for those seeking deeper involvement [25,35].

While the primary focus of this paper centers around the clinical barriers to the adoption of AI/ML in radiology, it is crucial to recognize and address additional aspects, such as technological/technical limitations, ethical considerations, and the imperative need for robust regulatory frameworks [36]. Among the practical barriers faced in the adoption of AI/ML in radiology, one notable issue is legal liability [36]. False negatives resulting in missed diagnoses or false positives potentially leading to unnecessary investigations/procedures can lead to detrimental outcomes for patients. To overcome this barrier,

it is essential to establish a clear framework for assigning responsibility in case of errors, ensuring that patients are not unfairly burdened with the consequences of such mistakes. Ethical considerations also play a significant role in the development and adoption of AI/ML in radiology, particularly concerning patient consent, data anonymization, and minimizing bias in the data used to develop these applications [36]; these topics are critical despite being beyond the scope of this review.

In addition, establishing comprehensive regulatory frameworks is essential to address concerns related to data privacy, patient safety, and the responsible deployment of AI into radiology and healthcare as a whole. Indeed, the absence of existing regulatory frameworks and guidelines from governing bodies currently presents a significant hurdle in the introduction and acceptance of AI into large healthcare institutions [37]. It is important to note that, although the current lack of regulatory approval is seen as a major barrier to adoption, this may change if more algorithms are approved or if the FDA updates its regulatory framework for modifications to AI- and ML-based software [37]. The FDA has already recognized the transformative potential of AI/ML in the software and medical device industry and, as a result, in April 2019, it published a discussion paper outlining a potential foundational approach to premarket review for AI- and ML-based software modifications [32].

Despite the above barriers, it is vital to acknowledge the immense potential of AI/ML to revolutionize patient care. Overcoming these challenges will require collaboration among various stakeholders, including radiologists, patients, healthcare organizations, regulatory bodies, and technology developers. Through collective efforts, we are optimistic that it is possible to navigate the barriers, harness the benefits of AI/ML, and ensure its responsible and impactful integration into radiology practice.

Outside of radiology, AI/ML has also made substantial advances in other medical fields, enabling achievements such as robotic surgical systems that mimic surgeons' movements and accurately identify tumor pathologies [38,39]. AI/ML technologies can also support physicians and other healthcare providers in a wide range of areas through direct patient care, including the transcription of notes and medical records, organizing patient information, reducing administrative tasks, and facilitating remote patient care. Despite the bright future of AI in different healthcare specialties, significant work needs to be performed for these innovations to be successfully integrated.

*Limitations and Future Research*

Although there has been an explosion of interest in research on AI in radiology, this scoping review identified a gap in AI literature related to barriers and enablers associated with implementing AI in radiology clinical practice. It found that few studies have been designed specifically to identify the attitudes of radiologists and clinicians towards the adoption of AI innovations. The small number of relevant studies limits the conclusions that can be made in this review, particularly given that it is challenging to directly compare the findings of qualitative and quantitative studies with different methods.

To address these gaps, further research should be conducted to explore barriers and enablers associated with the implementation of AI from the perspectives of physicians and other healthcare professionals. This will provide valuable insights into the experiences, concerns, and expectations of different professional groups, fostering a better understanding and adoption of AI technologies. Moreover, exploring patients' perspectives on the utilization of AI in healthcare will offer valuable insights into their acceptance, trust, and concerns, contributing to the patient-centered implementation of AI technologies.

Additionally, future research should extend beyond radiology and investigate the evolving role of AI in other medical specialties, like surgery and pathology. This broader investigation will help to identify potential barriers and facilitators in different domains of medicine, allowing radiologists to benefit from lessons learned in other specialties and leading to a more comprehensive understanding of AI implementation in healthcare.

Furthermore, it is crucial to address the accompanying concerns related to AI applications in the healthcare system, including technological limitations, ethical considerations, and the necessity for robust regulatory frameworks. Research efforts should prioritize the ethical development and use of AI in healthcare by examining aspects such as privacy, data security, and algorithmic biases.

## 5. Conclusions

With the recent expansion of AI applications in radiology, understanding the barriers and enablers associated with the implementation of AI into clinical practice is essential. A critical barrier is the uncertainty with regard to the added value AI solutions might bring to clinical practice, which causes low acceptance of AI applications among radiologists, though this should improve as more research evidence of the added benefit of AI applications in the clinical setting becomes available. Given the relative dearth of studies specifically investigating the barriers and enablers associated with the implementation of AI in radiology, further research into this area is important in order to have a more complete understanding of the issues likely to be pertinent as AI technologies inevitably become integrated into clinical practice.

**Supplementary Materials:** The following supporting information can be downloaded at: https://www.mdpi.com/article/10.3390/tomography9040115/s1, Table S1: Medline search strategy; Table S2: Embase search strategy.

**Author Contributions:** Conceptualization, F.A.E. and P.N.T.; methodology, F.A.E., M.A. and E.B.; software, F.A.E. and E.B.; validation, F.A.E., M.A., E.B., A.A. and P.N.T.; formal analysis, F.A.E., M.A. and E.B.; Investigation, F.A.E., M.A. and E.B.; resources, F.A.E., M.A. and E.B.; data curation, F.A.E. and M.A.; writing—original draft preparation, F.A.E., M.A. and E.B.; writing—review and editing, M.A., E.B. and P.N.T.; visualization, M.A.; supervision, P.N.T.; project administration, M.A., A.A. and P.N.T. All authors have read and agreed to the published version of the manuscript.

**Funding:** This research received no external funding.

**Institutional Review Board Statement:** Not applicable.

**Informed Consent Statement:** Not applicable.

**Data Availability Statement:** Medline and Embase databases.

**Conflicts of Interest:** The authors declare no conflict of interest.

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
