# Peer review of "Analyzing Barriers and Enablers for the Acceptance of Artificial Intelligence Innovations into Radiology Practice: A Scoping Review"

_tomography, doi:10.3390/tomography9040115_

Round 1
Reviewer 1 Report
I think the paper is well written and explains the barriers and challenges in mass AI adoption in clinical settings. However, my impression is that the authors have only looked at the non-quantifiable "feel of AI adoption by radiologists" as a barrier to AI adoption. There are many more technical factors as well, like FDA clearance and hurdles in building AI tools as outlines in the following paper: [2212.14177] Current State of Community-Driven Radiological AI Deployment in Medical Imaging (arxiv.org).
I think the authors should also address that in some capacity.
I think the paper is an attempt to understand the AI landscape for radiology and the authors do a good job summarizing a large number of survey articles. But I fail to see any suggestions from authors about addressing those challenges.
I will also like to see some discussion about issues/challenges they faced while deploying AI tools in their own capacity. There are quite a few similar survey articles that tackle the subject in discussion. I feel if the authors add a section on their own experience, it will make the paper better and standout.
Reviewer 2 Report
0. Abstract
- The conclusion section seems to belong to the results and advances in knowledge to conclusion.
1. Introduction
- There is repetition in the text. Reduce repetitive statements in the introduction to improve conciseness.
- The text flow needs improvement. Improve the coherence of the text by ensuring a logical progression of ideas without abrupt shifts.
- Enhance the clarity and scientific soundness of the writing.
- Emphasize the information on barriers and enablers.
2. Research Methods and study selection
2.1 Research methods
- It may be beneficial to include keywords such as "machine learning" and "deep learning" since they are commonly used in scientific articles as replacement of AI.
2.2 Study Selection
- Here you say that the 3 researchers independently evaluated the articles, and a fourth reviewer resolved the disagreement. Elaborate on this and explain why this is different that what you wrote in the abstract since you said that there were 2 independent evaluators, and a fourth reviewer to resolve the disagreement.
3. Results
- It is observed that several studies primarily present the perspectives of students rather than radiologists or clinicians, which deviates from the intended focus of this article as outlined in the objectives and title.
4. Discussion
- In general, the discussion section provides valuable insights; however, it could benefit from improvements in text flow to ensure a more coherent presentation of ideas.
Round 2
Reviewer 2 Report
I would like to thank the authors for addressing my previous comments. I see that they have put a lot of effort to improve the manuscript. However, I have some minor comments and one major one.
Minor:
- Lines 61-64: I strongly disagree with the referencing to CAD a technology without ability to learn. Although CAD terms are used previously to refer to the decision support systems that relies on if-then conditions or classical machine learning models, but currently and for the last few years it became a term that refers to even the new machine learning or deep learning-based decision support systems. CAD refers to the system which uses models and methods. These models and methods could be based on if-then, classical machine learning, modern machine learning, or/and deep learning methods/models. Therefore, I would suggest that you rephrase that and use a different term for old systems or be specific like “old-CAD systems” or something similar. Also make the changes through the manuscript.
- Lines 95-96: You may need to mention the medical students here as well.
- Table 1: It is kind of weird that you refer to continents (Europe, Africa, …etc) under the country column. Maybe you can replace “Country” with “Region” and then refer to regions/ continents instead of countries.
Major:
- Regarding including “machine learning” and maybe “deep learning” in the search. In lines 61-62 you say, “The terms AI and ML are commonly used interchangeably in medical imaging and will be used as such in this paper”. I agree with that. However, in your replay to my previous review you say, “While machine learning is an integral component that complements artificial intelligence, our focus primarily revolved around papers encompassing artificial intelligence, resulting in the exclusion of papers solely focused on machine learning.” This does not make sense to me. AI and ML are exchangeable terms, and in modern days when we refer to AI, we mean ML or deep learning, not the large umbrella of AI. So, it makes sense that many papers will publish using titles/terms include machine learning or deep learning instead of AI. I strongly recommend that you reconsider this and if you still insist on not including them at least you clearly state that as an exclusion criteria pointing the number of studies were excluded due to that and justify that in your manuscript.
